# All-in-One: Prompt-Driven Mixture of Hallucination-Aware Experts for Universal Anomaly Detection Across Multi-Modal Multi-Organ Medical Images

## Abstract

Unsupervised anomaly detection in medical images facilitates practical clinical adoption by identifying abnormalities without relying on scarce and costly annotated data. However, prior works have predominantly focused on specialized models for individual organs and modalities, impeding knowledge transfer and scalable deployment. In this paper, we investigate a task of universal anomaly detection guided by natural language prompts. We propose a prompt-driven mixture of experts framework that detects anomalies across multiple organs and modalities within a single network. Specifically, our method comprises encoders for vision and text, a routing network, and a mixture of hallucination-minimized expert decoders. An image and a prompt describing the organ and modality are fed to the encoders. The routing network then selects specialized yet collaborative expert decoders to analyze the image. We observe that anomaly detection models often erroneously identify normal image regions as anomalous, a phenomenon we term "hallucinatory anomaly". To address this issue, we design hallucination-aware experts that produce improved anomaly maps by jointly learning reconstruction and minimizing these false positives. For comprehensive evaluation, we curate a diverse dataset of 12,153 images spanning 5 modalities and 4 organs. Extensive experiments demonstrate state-of-the-art anomaly detection performance in this universal setting. Moreover, the natural language conditioning enables interpretability and user interaction. The code and data will be made publicly available.

## 1 Introduction

Deep learning has achieved remarkable success across a variety of computer vision tasks, yet its application to medical image analysis remains constrained by the need for sizable annotated datasets. Obtaining annotations for abnormal images proves particularly challenging, especially for rare or novel conditions (Tschuchnig & Gadermayr, 2022). In contrast, collections of normal medical images can be accumulated with relative ease. This disparity motivates anomaly detection in medical images—identifying abnormalities without reliance on annotated anomalous data during training.

Prior works (Shvetsova et al., 2021; Schlegl et al., 2017; Han et al., 2021a; Schlegl et al., 2019b; Jiang et al., 2019) have explored generative models, including autoencoders and generative adversarial networks (GANs), for unsupervised anomaly detection. These models are trained to learn feature representations using only normal images. At test time, anomalies are identified as regions that the models fail to reconstruct properly, by comparing input and reconstructed images in pixel space. Recent approaches utilize memory banks (Gong et al., 2019; Park et al., 2020), normalizing flows (Rudolph et al., 2021; Yu et al., 2021; Gudovskiy et al., 2022), self-supervised learning, and knowledge distillation to achieve stronger image-level anomaly detection performance. Despite promising results, these works have largely focused on training specialized models for individual organs and modalities. This methodology overlooks potential similarities across organs and modalities, hinders knowledge transfer, impedes scalable anomaly detection, and leads to fragmented research efforts.

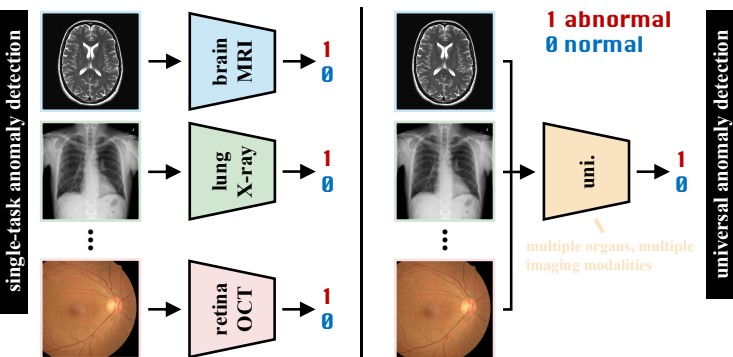

Figure 1: Illustration of single-task and universal medical anomaly detection models.

Universal anomaly detection is therefore desired (see Figure 1). Recently, You et al. (2022) pioneer a unified framework capable of detecting multiple industrial anomalies, sparking subsequent research in this direction (Lu et al., 2023; Zhao, 2023; Yao et al., 2024). Zhang et al. (2023) develop a single network for anomaly detection in medical images across two organs (lung and liver) and two modalities (CT and X-ray). However, these models rely solely on bottom-up processing to identify the organ and modality associated with each input image. In contrast, prompt-based approaches enable users to specify which anatomical structure to analyze in a given image, directing the model's attention in a top-down manner. We argue that conditioning universal anomaly detectors on natural language prompts confers considerable advantages in terms of model interpretability and user interaction. These benefits render prompt-guided universal anomaly detection more suitable for practical clinical adoption.

In this paper, we investigate the task of universal anomaly detection from natural language expressions, which leverages text to guide multi-modal, multi-organ anomaly detection in a single model. To address this task, we propose a prompt-driven mixture of experts framework comprising four key components: a vision encoder, a text encoder, a routing network, and a mixture of hallucination-minimized expert decoders. Specifically, an input image and an accompanying text prompt encapsulating organ and modality information are encoded by the vision and text encoders, respectively. The resulting representations are then combined and fed to the routing network to select decoder subnetworks (which we call experts) best suited for the given input. This design facilitates both cooperation, through shared representation learning, and specialization, by matching experts with specific tasks. Consequently, multi-modal, multi-organ images can be dynamically routed to appropriate sub-expert networks based on text prompts. The use of prompts maximizes the mutual information between experts and

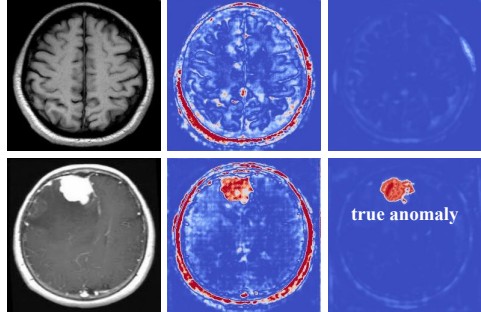

Figure 2: Illustration of hallucinatory anomalies. Top row: normal example; bottom row: abnormal example. Red intensity correlates with anomaly magnitude. Middle column: anomaly maps generated by an autoencoder-based anomaly detection model. Note that hallucinatory anomalies appear in both normal and abnormal images. Rightmost column: anomaly maps produced by our method, effectively eliminating these hallucinatory anomalies.

tasks, inducing a strong dependency where each task associates heavily with a small set of experts. For the experts, we devise a hallucination-aware decoder architecture that outputs a pixel-wise hallucinatory anomaly estimate in addition to a reconstructed image. We define "hallucinatory anomalies" as normal image regions that are erroneously identified as anomalous by an anomaly detection model (cf. Figure 2). By normalizing reconstruction errors with the predicted hallucinatory anomaly estimates, we obtain an abnormality score map that amplifies true anomalies while suppressing hallucinatory anomalies in normal regions.

Our key contributions are three-fold:

- This work presents an effort towards prompt-guided multi-organ, multi-modal anomaly detection within a single network. We collect a large-scale dataset spanning 12,153 images across 5 imaging modalities (X-ray, MRI, OCT, ultrasound, and CT) and 4 anatomical structures (lung, brain/head, retina, and breast).

- We propose a novel mixture of experts framework for this task. Our model is capable of routing images to suitable hallucination-minimized expert decoders in a collaborative yet specialized manner based on text prompts.

- We benchmark our method against state-of-the-art universal and single-task anomaly detection models. Experimental results demonstrate the superiority of our framework.

## 2 RELATED WORK

**Single-Task Anomaly Detection.** Reconstruction-based methods have emerged as a prominent approach in unsupervised anomaly detection. Schlegl et al. (2017) pioneer the use of GANs for this purpose with AnoGAN, later introducing f-AnoGAN (Schlegl et al., 2019a), a faster variant employing an encoder to map images to a latent space. In addition, various autoencoder architectures are explored, including variational autoencoder (Zimmerer et al., 2018) and vector-quantized variational autoencoder (Naval Marimont & Tarroni, 2021). To address the overgeneralization problem, where abnormal images are reconstructed too accurately, Gong et al. (2019) and Park et al. (2020) introduce memory banks to store normal patterns for comparison during inference. Several works (Rudolph et al., 2021; Gudovskiy et al., 2022; Yu et al., 2021) leverage normalizing flows, enabling exact likelihood estimation for image modeling, and achieve good performance in anomaly detection. Self-supervised learning (Jing & Tian, 2021) has also been applied to anomaly detection, typically following two paradigms. One-stage approaches train models to detect synthetic anomalies and directly apply them to real abnormalities (Tan et al., 2021; Schlüter et al., 2022). Two-stage approaches first learn self-supervised representations on normal data, followed by constructing one-class classifiers (Li et al., 2021; Sohn et al., 2021). Recently, knowledge distillation from pre-trained models presents another promising approach for unsupervised anomaly detection (Salehi et al., 2021; Deng & Li, 2022; Batzner et al., 2024). In these methods, a student network distilled by a pre-trained teacher network on normal samples can only extract normal features, leading to discrepancies when anomalies are encountered during inference. Despite their successes, the above-mentioned approaches have largely focused on dataset-specific models, potentially overlooking cross-class similarities and becoming resource-intensive as the number of classes increases.

**Universal Anomaly Detection.** You et al. (2022) first formulate universal anomaly detection, proposing a Transformer-based feature reconstruction model using a layer-wise query decoder to model complex multi-class normal distributions. Lu et al. (2023) present a unified hierarchical vector quantized Transformer that quantizes visual features to better reconstruct normal patterns. Yao et al. (2024) propose inter-class Gaussian mixture modeling and intra-class mixed class centers learning for multi-class anomaly detection. Beyond detection, Zhao (2023) focuses on universal anomaly localization for industrial applications. In the medical domain, Zhang et al. (2023) develop a single network capable of detecting anomalies across two organs (lung and liver) and two imaging modalities (CT and X-ray). The existing methods rely solely on visual features to identify the organ and modality of each input image through a bottom-up fashion. We propose that introducing text prompts can guide a universal anomaly detection model's attention in a top-down manner, leading to improved performance.

**Mixture of Experts.** Originally introduced by Jacobs et al. (1991), the mixture of experts framework combines multiple sub-models for conditional computation. Its integration with large language models has yielded remarkable results in natural language understanding tasks, to name a few, machine translation (Shazeer et al., 2017) and open-domain question answering (Du et al., 2022; Artetxe et al., 2022). This success has inspired applications in computer vision. Riquelme et al. (2021) introduce vision mixture of experts, matching the performance of leading networks while reducing computational demands during inference in image classification. Hwang et al. (2023) develop Tutel, a scalable system design and implementation for mixture of experts with dynamic parallelism and pipelining. Chowdhury et al. (2023) present patch-level routing, dynamically allo-

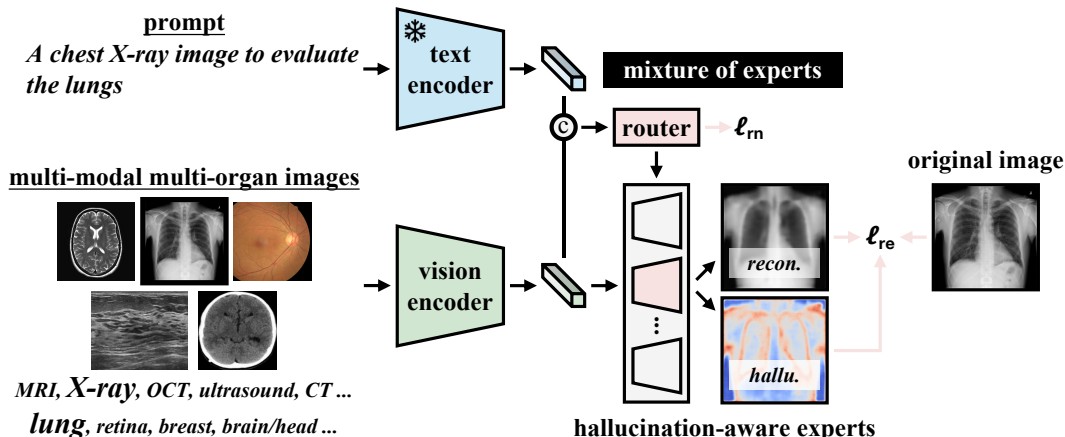

Figure 3: Proposed architecture for universal anomaly detection across multi-modal, multi-organ medical images.

cating image patches to experts through prioritized routing. Ye & Xu (2023) propose a multi-task mixture of experts model that enables learning multiple representative task-generic feature spaces and decoding task-specific features in a dynamic manner. Zhao et al. (2023a) leverage the mixture of experts architecture to learn from weak and noisy labels for detecting anomalies such as malware. Wang et al. (2024) make use of the mixture of experts framework to merge general knowledge from the segment anything model (SAM) (Kirillov et al., 2023) with domain-specific knowledge from task-specific fine-tuned models for volumetric medical image segmentation. In this paper, we propose a tailored mixture of experts model to address the hallucinatory anomaly problem.

## 3 METHODOLOGY

We assume that during training, only normal data are available, i.e., $\mathcal{D} = \{(\boldsymbol{x}_i, \boldsymbol{p}_i, y_i = 0)\}_{i=1}^{|\mathcal{D}|}$, where $y_i \in \{0, 1\}$ is a binary label indicating whether $\boldsymbol{x}_i$ is a normal ($y_i = 0$) or abnormal ($y_i = 1$) image, and $\boldsymbol{p}_i$ corresponds to the prompt of $\boldsymbol{x}_i$. We train the proposed model (cf. Fig. 3) using $\mathcal{D}$. In what follows, we delve into each part of our method.

### 3.1 SHARED VISION ENCODER

The shared vision encoder maps input images into a common latent feature space that is accessed by multiple decoder experts. Architecturally, the vision encoder consists of several convolutional blocks, each comprising Conv-BN-ReLU layers, followed by two fully connected layers. Formally, given an input image $\boldsymbol{x}$, the vision encoder $f(\cdot)$ produces a visual feature vector $\boldsymbol{v} = f(\boldsymbol{x})$.

### 3.2 TASK-SPECIFIC PROMPT ENCODER

Incorporating critical priors in a top-down manner is vital for directing the model's attention to appropriate tasks and image regions. To this end, we design a task-specific prompt encoder that takes textual prompts as input and generates task-specific feature representations. Specifically, we leverage the text encoder from CLIP (Radford et al., 2021) to extract linguistic features, as it is pretrained on a massive corpus of image-text pairs. We then apply a fully connected layer to condense the text features. Formally, given a prompt $\boldsymbol{p}$, the output of the prompt encoder is $\boldsymbol{\tau} = g(\boldsymbol{p})$, where $g(\cdot)$ denotes the prompt encoder.

## 3.3 MIXTURE OF EXPERTS

### 3.3.1 ROUTING NETWORK

To adaptively control the contribution of each expert, we make use of a routing network. The concatenation of textual and visual features is input to this router to produce expert selections specialized for each task. We implement the routing network as follows:

$$s = \text{TopK}(\text{softmax}([\boldsymbol{v}, \boldsymbol{\tau}]\boldsymbol{W} + \boldsymbol{b})), \tag{1}$$

where $\boldsymbol{s} \in \mathbb{R}^N$, and $N$ is the number of experts. The $\text{TopK}$ operator forces only $K$ experts $(K \leq N)$ to be used and skips the others. $\boldsymbol{W}$ and $\boldsymbol{b}$ are learnable parameters. Subsequently, the router chooses the most task-relevant experts and aggregates their representations for different anomaly detection tasks. Furthermore, we introduce a loss function that drives our model to learn optimal expert-task matchings (cf. Section 3.4).

### 3.3.2 HALLUCINATION-AWARE EXPERTS

To address the critical issue of hallucinatory anomalies in anomaly detection, we design hallucination-minimized experts. Specifically, for the $k$-th expert, we set two output channels in its ultimate block, denoted as $\boldsymbol{\mu}^k$ and $\boldsymbol{\sigma}^k$: the former for reconstructing the input image, and the latter for predicting per-pixel hallucination propensity. Our motivation stems from the following observation: Anomaly detectors often produce high reconstruction errors not only in abnormal regions but also along boundaries of normal areas. These boundary-induced errors generally lead to misidentification of anomalies, resulting in what term hallucinatory anomalies (Figure 2). Therefore, we would like to utilize hallucination quantification to rectify erroneous boundary detections and thus improve the localization of truly anomalous regions. To this end, we devise such a hallucination-aware decoder architecture and a corresponding loss function (see Section 3.4). Finally, the output of the expert squad is a weighted sum of reconstructions from individual decoder experts, with weights calculated by the routing network, conditioned on the input image and prompt. This process is expressed as:

$$\hat{\boldsymbol{x}} = \sum_{k=1}^{N} s^k \boldsymbol{\mu}^k. \tag{2}$$

Similarly, we obtain

$$\boldsymbol{u} = \sum_{k=1}^{N} s^k \boldsymbol{\sigma}^k. \tag{3}$$

### 3.4 TRAINING OBJECTIVES

In each iteration, we sample a data batch $\mathcal{B} = \{(\boldsymbol{x}_i, \boldsymbol{p}_i)\}_{i=1}^{|\mathcal{B}|}$. Let $\boldsymbol{c}_i$ denote the category of organ and modality for the $i$-th image. First, we optimize the matching between experts and tasks by minimizing the discrepancy between the router's prediction and $\boldsymbol{c}_i$:

$$\mathcal{L}_{\text{rn}} = -\frac{1}{|\mathcal{B}|} \sum_{i=1}^{|\mathcal{B}|} \boldsymbol{c}_i \log(\boldsymbol{s}_i). \tag{4}$$

Next, we optimize the reconstruction process while accounting for potential hallucinatory anomalies, which is a key contribution of our hallucination-minimized experts:

$$\mathcal{L}_{\text{re}} = \frac{1}{|\mathcal{B}|M} \sum_{i=1}^{|\mathcal{B}|} \sum_{j=1}^{N} ((\boldsymbol{x}_{i,j} - \hat{\boldsymbol{x}}_{i,j})^2 e^{-\boldsymbol{u}_{i,j}^2} + \boldsymbol{u}_{i,j}^2), \tag{5}$$

where $j$ is a spatial index, and $M$ indicates the number of pixels. During training on **normal** images, the first loss term discourages our model from predicting very small hallucination scores for pixels with high reconstruction errors, as reducing hallucination propensity amplifies the impact of already large reconstruction errors. Conversely, the second loss term drives hallucination scores in other regions to be small. Therefore, the two loss terms jointly optimize our model to estimate low

hallucination scores in regions with accurate reconstructions, while predicting relatively high scores near boundaries in normal images. Finally, the loss in our model is defined as:

$$\mathcal{L} = \alpha \mathcal{L}_{\mathrm{rn}} + \beta \mathcal{L}_{\mathrm{re}}, \tag{6}$$

where $\alpha$ and $\beta$ are two coefficients balancing the two loss terms.

### 3.5 Anomaly Scoring

During inference on a test image, the mean of pixel-wise reconstruction errors has been widely adopted as an anomaly score for the image. In this work, we use an anomaly score calculation method based on our hallucination-minimized expert model. We leverage the first term in Eq. 5 to compute the anomaly score. Specifically, $(\boldsymbol{x}_i - \hat{\boldsymbol{x}}_i)$ represents the reconstruction error map generated by our model, while $\boldsymbol{u}_i$ denotes the corresponding hallucination quantification map. Through training on normal data, we have developed a model capable of estimating high hallucination values when significant reconstruction errors occur in normal regions. Consequently, during inference on both normal and abnormal images, reconstruction errors in normal regions can be effectively rectified using the first term in Eq. 5. This results in an anomaly map that accurately localizes true anomalous regions. The final anomaly score for the test image is computed as the mean value of this anomaly map.

## 4 Experiments

### 4.1 Data

To evaluate our approach, we compile a comprehensive, multi-modal, multi-organ universal anomaly detection dataset by integrating five medical imaging datasets: the RSNA Pneumonia Detection Challenge dataset[1], the Brain Tumor MRI dataset[2], the Large-scale Attention-based Glaucoma (LAG) dataset (Li et al., 2019), the Breast Ultrasound Images (BUSI) dataset (Al-Dhabyani et al., 2020), and the HeadCT dataset[3].

**RSNA:** This chest X-ray dataset contains 8,851 normal and 6,012 lung opacity images. Following Cai et al. (2022), we use 3,851 normal images for training and a balanced test set of 1,000 normal and 1,000 abnormal images.

**Brain Tumor:** This dataset consists of 2,000 MRI slices without tumors, 1,621 with gliomas, and 1,645 with meningiomas. We categorize glioma and meningioma slices as anomalies. The normal instances are sourced from Br35H5 and Saleh et al. (2020), while the anomalous cases are from Saleh et al. (2020) and Cheng et al. (2015). In line with Cai et al. (2022), our experimental setup includes 1,000 normal slices for training and a test set of 600 normal and 600 abnormal slices (equally split between glioma and meningioma).

**LAG:** This dataset comprises 3,143 normal retinal fundus images and 1,711 abnormal retinal fundus images with glaucoma. Following Cai et al. (2022), we use 1,500 normal images as training samples and 811 normal and 811 abnormal images as test examples.

**BUSI:** The dataset includes 133 normal breast ultrasound images, 437 images with benign nodules, and 210 images with malignant nodules. We use 99 normal images for training, with the remaining images used for evaluation.

**HeadCT:** This dataset comprises 100 normal head CT slices and 100 slices with hemorrhage. We divide these images into two groups: 90 normal images for training and 10 normal with 100 abnormal images for testing.

---

[1]https://www.kaggle.com/c/rsna-pneumonia-detection-challenge
[2]https://www.kaggle.com/datasets/masoudnickparvar/
brain-tumor-mri-dataset
[3]https://www.kaggle.com/datasets/felipekitamura/head-ct-hemorrhage

| | RSNA | | | Brain Tumor | | | LAG | | |
|---|---|---|---|---|---|---|---|---|---|
| | AUC | F1 | ACC | AUC | F1 | ACC | AUC | F1 | ACC |
| Single-Task Anomaly Detection | | | | | | | | | |
| AE | 68.33 | 67.85 | 52.90 | 80.88 | 84.79 | 82.33 | 78.19 | 74.91 | 72.87 |
| MemAE | 68.65 | 67.95 | 53.45 | 77.44 | 79.92 | 76.33 | 80.78 | 76.16 | 74.72 |
| CFLOW-AD | 70.26 | 70.20 | 62.05 | 36.35 | 66.67 | 50.00 | 43.38 | 66.67 | 50.00 |
| FastFlow | 76.00 | 73.68 | 67.95 | 85.62 | 80.41 | 77.58 | 77.40 | 75.42 | 71.52 |
| GAN Ensemble | 82.10 | 75.30 | 74.30 | 66.60 | 68.40 | 64.00 | 61.30 | 67.10 | 52.50 |
| CutPaste | 55.82 | 66.69 | 50.05 | 58.45 | 67.61 | 54.25 | 53.86 | 66.83 | 50.62 |
| NSA | 82.13 | 75.87 | 74.30 | 83.20 | 79.00 | 76.17 | 72.67 | 73.57 | 67.57 |
| MorphAEus | 80.87 | 75.74 | 72.80 | 64.68 | 70.43 | 60.67 | 79.03 | 78.89 | 74.17 |
| SQUID | 70.38 | 72.40 | 65.95 | 41.33 | 66.67 | 50.00 | 55.76 | 66.89 | 52.03 |
| EfficientAD | 74.88 | 73.12 | 68.20 | 78.41 | 76.20 | 72.00 | 73.63 | 72.93 | 68.74 |
| Universal Anomaly Detection | | | | | | | | | |
| UniAD | 80.05 | 72.05 | 74.44 | 70.35 | 71.51 | 60.42 | 70.55 | 71.45 | 63.44 |
| HVQ-Trans | 82.72 | 76.21 | 73.75 | 82.14 | 76.54 | 71.50 | 74.62 | 74.38 | 66.52 |
| MADDR | 82.56 | 77.11 | 75.00 | 87.07 | 82.57 | 83.42 | 81.42 | 77.42 | 74.85 |
| HGAD | 78.84 | 75.66 | 72.40 | 90.07 | 85.67 | 84.83 | 76.85 | 77.79 | 72.01 |
| Ours | **83.51** | **78.54** | **75.95** | **93.48** | **89.51** | **89.00** | **84.77** | **80.66** | **78.48** |

| | BUSI | | | HeadCT | | | MEAN | | |
|---|---|---|---|---|---|---|---|---|---|
| | AUC | F1 | ACC | AUC | F1 | ACC | AUC | F1 | ACC |
| Single-Task Anomaly Detection | | | | | | | | | |
| AE | **88.78** | 98.18 | 96.48 | 89.10 | 96.04 | 92.73 | 81.06 | 84.35 | 79.46 |
| MemAE | 85.93 | 98.32 | 96.77 | 85.70 | 96.04 | 92.73 | 79.70 | 83.68 | 78.80 |
| CFLOW-AD | 73.69 | 97.51 | 95.15 | 83.10 | 95.65 | 91.82 | 61.36 | 79.34 | 69.80 |
| FastFlow | 82.58 | 97.73 | 95.59 | 72.40 | 96.15 | 92.73 | 78.80 | 84.68 | 81.07 |
| GAN Ensemble | 44.80 | 97.40 | 95.00 | 40.20 | 95.20 | 90.90 | 59.00 | 80.68 | 75.34 |
| CutPaste | 57.26 | 97.44 | 95.01 | 57.50 | 95.24 | 90.91 | 56.58 | 78.76 | 68.17 |
| NSA | 74.47 | 87.17 | 96.48 | 93.03 | 95.24 | 90.91 | 81.10 | 82.17 | 81.09 |
| MorphAEus | 72.83 | 97.51 | 95.15 | 48.80 | 95.24 | 90.91 | 69.24 | 83.56 | 78.74 |
| SQUID | 66.95 | 97.51 | 95.15 | 75.60 | 95.69 | 91.82 | 62.00 | 79.83 | 70.99 |
| EfficientAD | 88.26 | 98.32 | 96.77 | 74.90 | 95.69 | 91.82 | 78.02 | 83.25 | 79.51 |
| Universal Anomaly Detection | | | | | | | | | |
| UniAD | 81.10 | 97.44 | 95.01 | 82.50 | 95.69 | 91.82 | 76.91 | 81.63 | 77.02 |
| HVQ-Trans | 85.48 | 98.03 | 96.18 | 90.90 | 95.69 | 91.82 | 83.17 | 84.17 | 79.95 |
| MADDR | 85.70 | 94.98 | 90.75 | 86.00 | 60.14 | 48.18 | 84.55 | 78.44 | 74.44 |
| HGAD | 75.66 | 97.96 | 96.04 | 72.20 | 96.15 | 92.73 | 78.72 | 86.65 | 83.60 |
| Ours | 87.22 | **98.63** | **97.36** | **91.60** | **98.02** | **96.36** | **88.12** | **89.07** | **87.43** |

Table 1: Quantitative comparison of our model against other single-task and universal anomaly detection methods on five datasets. Performance is measured by AUC, F1 score, and ACC. The best results for each dataset and metric are highlighted in **bold**.

## 4.2 EVALUATION METRICS

Given that unsupervised anomaly detection methods typically generate continuous-valued predictions, we primarily use the area under a receiver operating characteristic (ROC) curve (AUC) as our evaluation metric due to its threshold-independent nature. Additionally, we report F1 score and accuracy. For these metrics, we determine the optimal threshold based on the best F1 score, following the approach of Zhao et al. (2023b).

## 4.3 IMPLEMENTATION DETAILS

All experiments are conducted using PyTorch on a single NVIDIA RTX 3090Ti GPU. We preprocess all images by resizing them to $256 \times 256$ pixels and train for 250 epochs using the Adam optimizer

| RSNA | Brain Tumor | LAG | BUSI | HeadCT |
| --- | --- | --- | --- | --- |

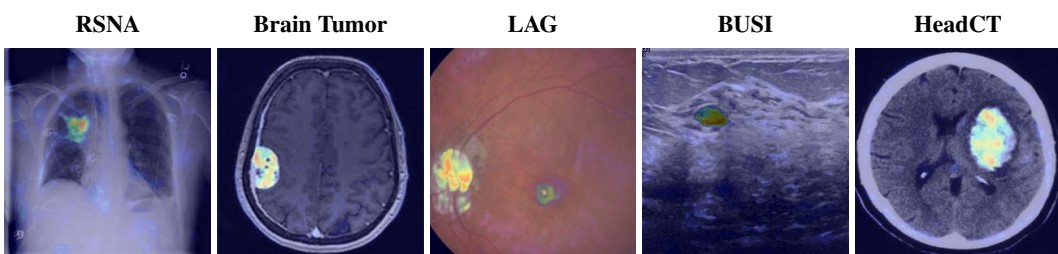

Figure 4: Visualization of exemplar anomaly maps generated by the proposed model.

with a learning rate of 5e-4 and a batch size of 64. The shared encoder contains four convolutional layers (each with a $4 \times 4$ convolution, stride 2), whose channel sizes are 16-32-64-64, followed by two fully connected layers with output sizes of 2048 and 16, respectively. Each decoder consists of four deconvolutional layers with the same kernel size and stride as the encoder, and the channel sizes are set to 64-32-16-2. All layers except the output layer are followed by batch normalization (BN) and ReLU. The routing network consists of a fully connected layer, the output size of which matches the number of decoders. For competing methods, we utilize their publicly available codes and adhere to their default training configurations.

## 4.4 COMPARISON WITH STATE-OF-THE-ART METHODS

We evaluate our proposed method against state-of-the-art approaches, including both single-task and universal anomaly detection models, across all datasets. The competing single-task models include AE, MemAE (Gong et al., 2019), FastFlow (Yu et al., 2021), GAN Ensemble (Han et al., 2021b), CutPaste (Li et al., 2021), CFLOW-AD (Gudovskiy et al., 2022), NSA (Schlüter et al., 2022), SQUID (Xiang et al., 2023), MorphAEus (Bercea et al., 2023), and EfficientAD (Batzner et al., 2024). For universal models, we compare against UniAD (You et al., 2022), HVQ-Trans (Lu et al., 2023), MADDR (Zhang et al., 2023), and HGAD (Yao et al., 2024).

Table 1 presents the comparative results. Our approach achieves the highest F1 scores across all five datasets, surpassing the best single-task models by 2.67%, 4.72%, 1.77%, 0.31%, and 1.87% on RSNA, BrainTumor, LAG, BUSI, and HeadCT, respectively. We also attain the highest accuracies across all datasets. In terms of AUC, our model outperforms the best single-task anomaly detection models on four datasets (RSNA: 1.38%, BrainTumor: 7.86%, LAG: 3.99%, HeadCT: 2.5%), while trailing the top performer on BUSI by 1.56%. Overall, our approach achieves the best average AUC, F1 score, and accuracy across the five datasets, leading the best single-task model by 7.06%, 4.39%, and 6.36%, respectively.

Furthermore, our framework consistently outperforms competing universal anomaly detection models. Compared to MADDR (Zhang et al., 2023), we achieve better average AUC (+3.56%), F1 score (+10.63%), and accuracy (+12.77%). Against HGAD (Yao et al., 2024), our approach demonstrates average improvements of 9.39%, 2.43%, and 3.82% in AUC, F1, and accuracy, respectively.

For qualitative analysis, we demonstrate our method's anomaly localization capability through example anomaly maps in Figure 4.

## 4.5 DISCUSSION

To provide insights into key components, we analyze our framework from the following four perspectives.

### 4.5.1 EXPERTS WITH VS. WITHOUT HALLUCINATION QUANTIFICATION

Compared to the model lacking hallucination quantification, our approach improves average AUC, F1 score, and accuracy by 7.27%, 5.06%, and 6.44% across datasets, respectively. Detailed gains for each dataset and evaluation metric are provided in the Table 2. Figure 5 shows anomaly score distributions, indicating discriminative power. Less overlap between normal and abnormal histograms

| HQ | TP | RSNA | | | Brain Tumor | | | LAG | | |
|---|---|---|---|---|---|---|---|---|---|---|
| | | AUC | F1 | ACC | AUC | F1 | ACC | AUC | F1 | ACC |
| - | ✓ | 67.10 | 67.82 | 52.75 | 76.71 | 80.11 | 76.25 | 77.09 | 73.38 | 69.54 |
| ✓ | - | 82.29 | 77.55 | 75.25 | 84.65 | 80.28 | 79.17 | 82.22 | 77.96 | 74.91 |
| ✓ | ✓ | **83.51** | **78.54** | **75.95** | **93.48** | **89.51** | **89.00** | **84.77** | **80.66** | **78.48** |

| | | BUSI | | | HeadCT | | | MEAN | | |
|---|---|---|---|---|---|---|---|---|---|---|
| | | AUC | F1 | ACC | AUC | F1 | ACC | AUC | F1 | ACC |
| - | ✓ | 87.16 | 98.33 | **97.36** | 91.30 | 97.06 | 94.55 | 79.87 | 83.34 | 78.09 |
| ✓ | - | 83.56 | 98.55 | 97.21 | 91.10 | 97.03 | 94.55 | 84.76 | 86.27 | 84.22 |
| ✓ | ✓ | **87.22** | **98.63** | **97.36** | **91.60** | **98.02** | **96.36** | **88.12** | **89.07** | **87.43** |

Table 2: Ablation study quantifying the impact of each component in the proposed method on all datasets. We report the performance of our full model, as well as variants with the following components ablated: hallucination quantification (HQ) and text prompting (TP).

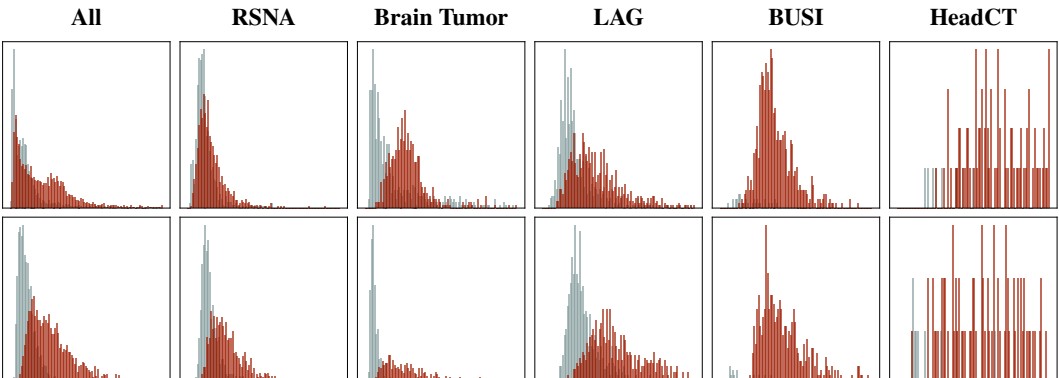

Figure 5: Distributions of anomaly scores for normal (grayish green) and abnormal (red) images in test sets for all datasets and for each dataset. Top: Anomaly score distributions obtained from our model without hallucination consideration. Bottom: Abnormality score distributions produced by our full model. x-axis: anomaly score from 0 to 1; y-axis: count.

enables stronger discrimination. The improved quantitative metrics and separation of distributions demonstrate the efficacy of our hallucination-minimized experts.

### 4.5.2 UNIVERSAL ANOMALY DETECTION WITH VS. WITHOUT TEXT PROMPTS

Under the same architecture (utilizing hallucination-minimized experts), incorporating prompts yields significant performance improvements: an average increase of 3.35% in AUC, 2.80% in F1 score, and 3.21% in accuracy. Detailed gains for each dataset and metric are presented in Table 2. Moreover, in Appendix B, we provide a visualization comparing the feature distribution obtained from our full model against that of the prompt-less method.

### 4.5.3 SELECTION OF HYPERPARAMETER $K$

Our experiments with varying values of $K$, as presented in Table 3, reveal that optimal performance is achieved when $K$ equals the number of experts ($K = N$). This finding suggests that the full ensemble of experts provides complementary knowledge or capabilities that are synergistically leveraged when all are active.

### 4.5.4 RELATION BETWEEN EXPERTS AND TASKS

Figure 6 visualizes the frequency of experts being selected for each task. The x-axis and y-axis represent experts and tasks, respectively. The visualization reveals sparsity and task-specificity in expert

| $K$ | RSNA | | | Brain Tumor | | | LAG | | |
|---|---|---|---|---|---|---|---|---|---|
| | AUC | F1 | ACC | AUC | F1 | ACC | AUC | F1 | ACC |
| 1 | 79.24 | 74.97 | 72.35 | 92.71 | 89.19 | 88.17 | 77.62 | 73.52 | 69.36 |
| 2 | 80.90 | 76.02 | 73.60 | 92.88 | 88.85 | 88.58 | 76.32 | 74.52 | 69.05 |
| 3 | 82.44 | 77.24 | 75.45 | 93.38 | **90.38** | **89.83** | 80.67 | 76.16 | 73.37 |
| 4 | 82.86 | 77.18 | 74.40 | 91.84 | 87.40 | 87.33 | 81.29 | 77.52 | 75.65 |
| 5 | **83.51** | **78.54** | **75.95** | 93.48 | 89.51 | 89.00 | **84.77** | **80.66** | **78.48** |

| $K$ | BUSI | | | HeadCT | | | MEAN | | |
|---|---|---|---|---|---|---|---|---|---|
| | AUC | F1 | ACC | AUC | F1 | ACC | AUC | F1 | ACC |
| 1 | 85.79 | 98.55 | 97.21 | 91.20 | **98.04** | **96.36** | 85.31 | 86.85 | 84.69 |
| 2 | 85.80 | 98.63 | 97.36 | **94.10** | 96.59 | 93.64 | 86.00 | 86.92 | 84.45 |
| 3 | **88.57** | **98.70** | **97.50** | 81.80 | 96.08 | 92.73 | 85.37 | 87.71 | 85.78 |
| 4 | 87.08 | 98.30 | 97.36 | 91.40 | 97.00 | 94.55 | 86.89 | 87.63 | 85.86 |
| 5 | 87.22 | 98.63 | 97.36 | 91.60 | 98.02 | **96.36** | **88.12** | **89.07** | **87.43** |

Table 3: Performance analysis for varying values of $K$ on diverse datasets, showing the impact of $K$ on model performance.

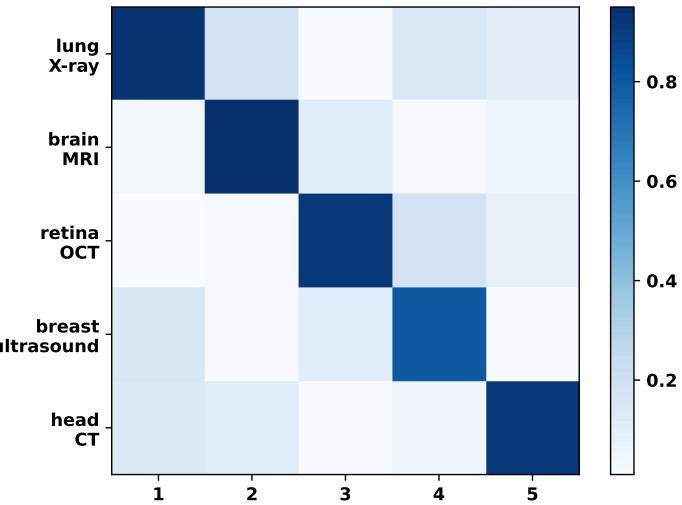

Figure 6: Heatmap visualization depicting the probability of each expert being selected for different tasks. The y-axis enumerates all tasks (organs and imaging modalities), while the x-axis represents the five experts in our model. Intensity of color correlates with selection frequency.

selection. For a given task, only a few experts are activated with high weights. In addition, similar tasks tend to activate the same experts, indicating task-specificity in the expert-task relationship.

## 5 CONCLUSION AND FUTURE WORK

In this paper, we propose a prompt-driven mixture of experts framework for universal anomaly detection across organs and modalities via natural language conditioning. Through encoders, routing networks, and the proposed hallucination-aware expert decoders, our method leverages both vision and text to detect anomalies within a single model. Extensive experiments on a diverse medical image dataset with over 12K images demonstrate state-of-the-art performance. The natural language prompts also enable model interpretability and user interaction. In the future, we plan to expand the framework to additional organs and modalities, investigate few-shot anomaly detection with limited normal images, and deploy the system in clinical settings to assist medical professionals.

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

# APPENDIX

## A   DATASET

| Task | Text Prompts |
|------|--------------|
| 1 | Chest X-ray image to evaluate the lungs. |
| 2 | Brain MRI slice. |
| 3 | Retinal fundus image. |
| 4 | Breast ultrasound image. |
| 5 | Head CT slice. |

Table 4: The text prompt corresponding to each task.

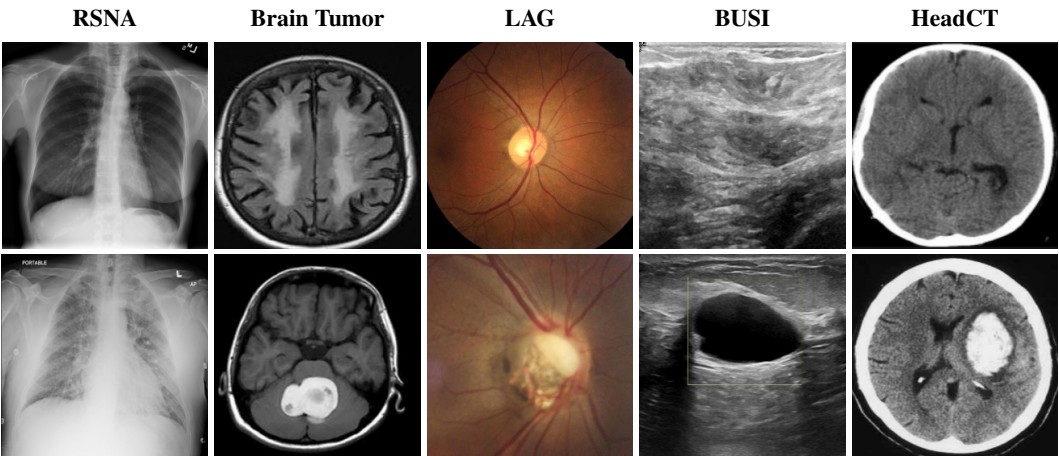

Figure 7: Sample test images from the dataset used in our study. Normal images are shown on the top, while abnormal images are displayed on the bottom.

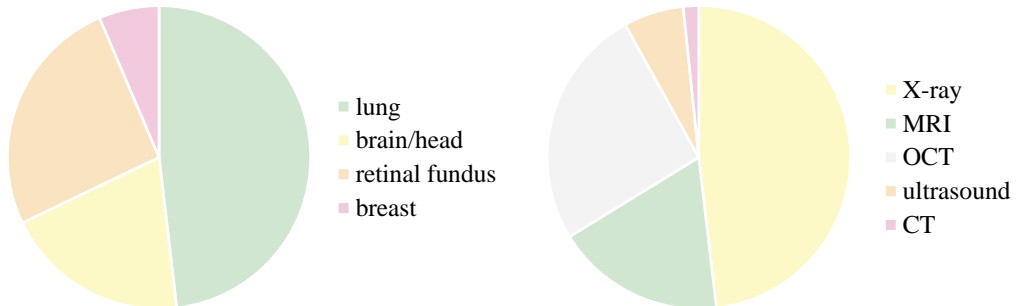

Figure 8: Organ (left) and modality (right) distributions in the dataset.

Figure 7 presents 10 example images from the anomaly detection tasks: 5 normal and 5 abnormal. The first row displays normal images, while the second row shows images containing anomalies. We visualize the distribution of organs and imaging modalities across our entire dataset in Figure 8. Furthermore, Table 4 presents the corresponding text prompts for each task, providing a comprehensive overview of the language prompts used to guide our model across various tasks.

## B ADDITIONAL EXPERIMENTAL RESULTS

Figure 9 visualizes feature distributions, where the ablated, prompt-less model struggles to accurately differentiate between tasks. In contrast, our full model effectively separates the distributions, enhancing anomaly detection across medical modalities and organs while improving interpretability.

We demonstrate our method's capability to localize abnormal regions for different anomaly detection tasks. As illustrated in Figure 10, Figure 11, Figure 12, and Figure 13, reconstruction errors of two competing methods are relatively large at some normal region boundaries. In contrast, our approach significantly reduces reconstruction errors at these boundaries through our hallucination quantification mechanism, thereby accurately pinpointing true abnormal regions.

In these figures, the first and second rows present normal examples, while the third and fourth rows show abnormal examples.

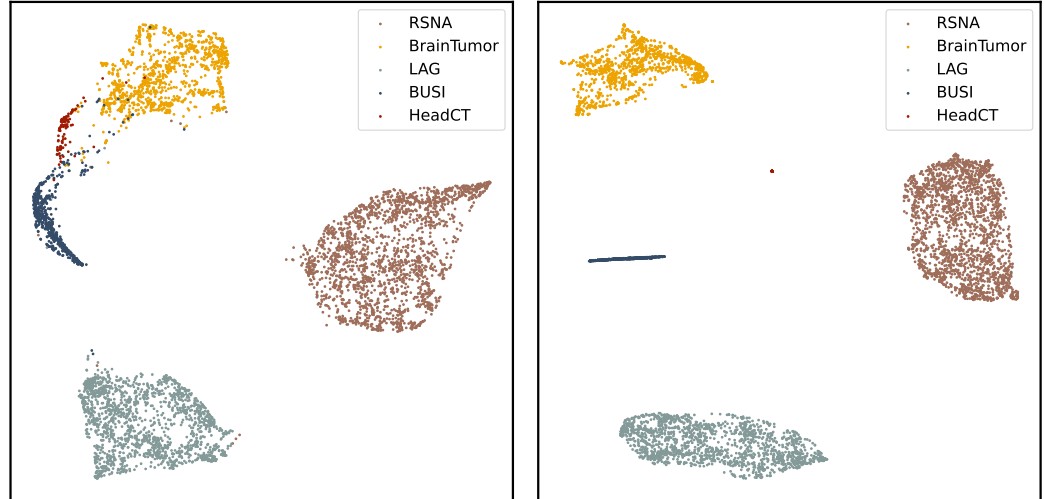

Figure 9: t-SNE visualization of feature distributions for five datasets. Left: Model without text prompts. Right: Full model with text prompts. Colors denote different datasets.

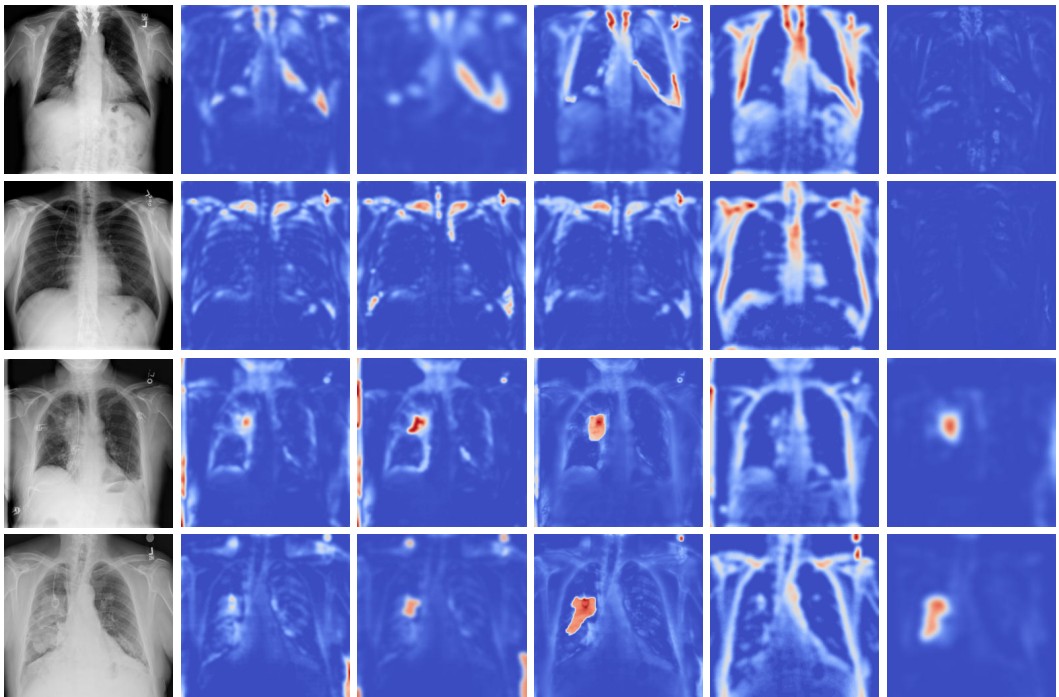

Figure 10: Anomaly localization visualization on the RSNA dataset. The columns are organized as follows: original images, anomaly maps generated by MemAE (Gong et al., 2019), anomaly maps generated by NSA (Schlüter et al., 2022), reconstruction error maps obtained by our method, hallucination quantification maps obtained by our method, and final anomaly maps generated by integrating the reconstruction error and hallucination quantification maps.

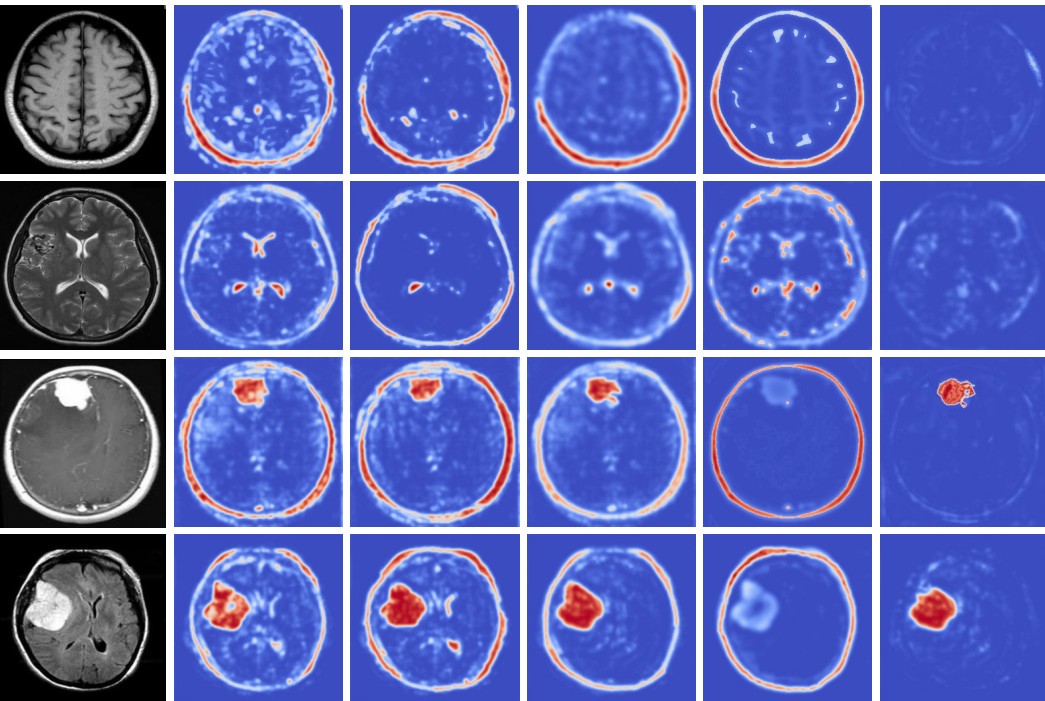

Figure 11: Anomaly localization visualization on the Brain Tumor dataset. The columns are organized as follows: original images, anomaly maps generated by MemAE (Gong et al., 2019), anomaly maps generated by NSA (Schlüter et al., 2022), reconstruction error maps obtained by our method, hallucination quantification maps obtained by our method, and final anomaly maps generated by integrating the reconstruction error and hallucination quantification maps.

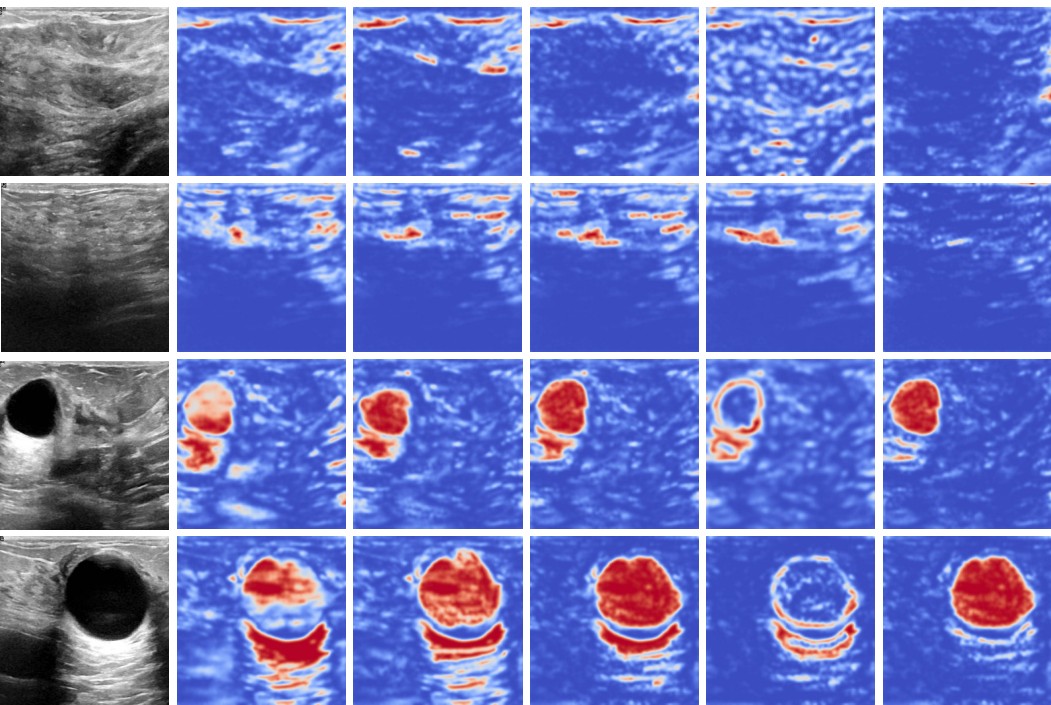

Figure 12: Anomaly localization visualization on the BUSI dataset. The columns are organized as follows: original images, anomaly maps generated by MemAE (Gong et al., 2019), anomaly maps generated by NSA (Schlüter et al., 2022), reconstruction error maps obtained by our method, hallucination quantification maps obtained by our method, and final anomaly maps generated by integrating the reconstruction error and hallucination quantification maps.

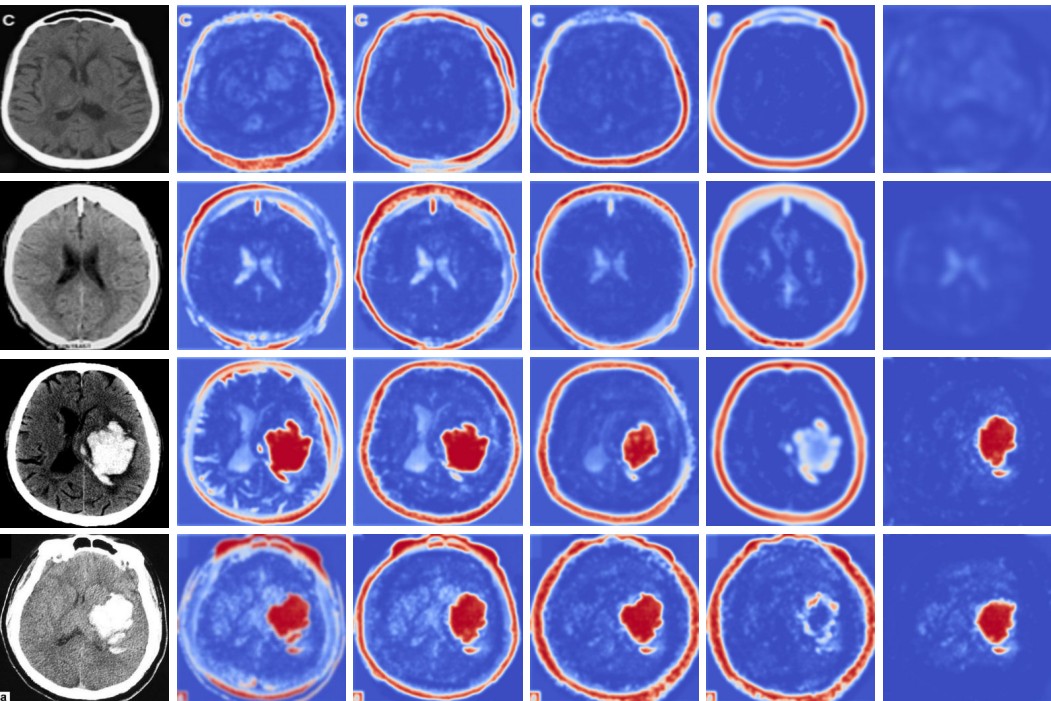

Figure 13: Anomaly localization visualization on the HeadCT dataset. The columns are organized as follows: original images, anomaly maps generated by MemAE (Gong et al., 2019), anomaly maps generated by NSA (Schlüter et al., 2022), reconstruction error maps obtained by our method, hallucination quantification maps obtained by our method, and final anomaly maps generated by integrating the reconstruction error and hallucination quantification maps.

