# OpenReview forum: "All-in-One: Prompt-Driven Mixture of Hallucination-Aware Experts for Universal Anomaly Detection Across Multi-Modal Multi-Organ Medical Images"
_ICLR.cc/2025/Conference — ICLR 2025 Conference Withdrawn Submission_

### Official Review · Reviewer_L2fA · 2024-11-01

**Soundness:** 2
**Presentation:** 3
**Contribution:** 2
**Rating:** 5
**Confidence:** 4

**Summary:**

The authors propose a prompt-driven mixture of experts (MoE) model that can detect anomalies across multiple organs and imaging modalities, addressing a gap where most existing models focus on specific organs or modalities.

**Strengths:**

1. The authors validate their method on a large dataset, spanning multiple organs and modalities, and demonstrate superior performance compared to other models.

2. The use of natural language prompts to guide anomaly detection is a practical idea.

**Weaknesses:**

1. There is limited discussion on the diversity and quality of the dataset, especially regarding rare conditions or imaging variations. And how the model might behave on noisy or imbalanced real-world medical datasets, where labels may be incomplete, inaccurate, or skewed.

2. While the paper claims to introduce a novel mixture of expert (MoE) architecture, the core idea of MoE is not new. The mixture of experts has been explored extensively in both natural language processing and computer vision. The author should clarify the differences and how it can work on the specific problem well.

3. The natural language prompts used in the experiments are relatively straightforward (e.g., “A chest X-ray image to evaluate the lungs”). The paper doesn’t address how the model would handle more complex or nuanced prompts.

4. While the hallucination-aware experts are designed to minimize false positives, the paper lacks a clear explanation of why and how these experts work effectively. The mechanism for detecting and quantifying hallucination anomalies is somewhat opaque, and there are no visual or empirical explanations of how these experts influence anomaly maps. Providing more visual examples of the hallucination-aware experts in action, along with a more intuitive explanation of their impact.

**Questions:**

1. How sensitive is the model to poorly formulated or ambiguous prompts? Does the model require highly specific prompts to perform well, or can it generalize to more vague inputs?

2. The paper focuses heavily on quantitative metrics (AUC, F1 score, ACC) without providing enough qualitative analysis of the model’s performance. For instance, how does the model handle borderline cases or ambiguous abnormalities?

---

### Official Review · Reviewer_KVxo · 2024-11-03

**Soundness:** 1
**Presentation:** 2
**Contribution:** 2
**Rating:** 3
**Confidence:** 4

**Summary:**

The paper proposes a novel framework for anomaly detection in medical images, aiming to detect abnormalities across multiple organs and imaging modalities within a single unified model. The authors address this limitation by introducing a prompt-driven mixture of experts framework that leverages natural language prompts to guide the anomaly detection process. They conduct experiments comparing their method with both single-task and universal anomaly detection models, demonstrating that their approach achieves SOTA performance. Additionally, they show that incorporating natural language prompts enhances interpretability and user interaction.

**Strengths:**

The paper tackles the challenging problem of universal anomaly detection across multiple organs and modalities using a single model, which is a significant step forward compared to traditional methods that focus on single-task settings.

The integration of natural language prompts to guide the model introduces a novel approach to anomaly detection.

Addressing the issue of "hallucinatory anomalies" is a contribution. By quantifying and minimizing these false positives, the method improves the accuracy of anomaly localization.

**Weaknesses:**

1. One of the advantages of MoE is to let the router automatically learn which experts should be utilized. But the proposed Lrn uses labels to point experts for a specific task. So it would be better to have an ablation study on how the router is learned and experts are selected without Lrn.

2. Moreover, using a specific text prompt and Lrn-based router selection mechanism might be a complex version of a framework with a shared vision encoder and keyword-based decoder selection system, and training them together. The advantages and potential of utilizing text prompts and mixture of experts are not well demonstrated or deeply analyzed in this paper.

3. The performance of the model may heavily rely on the quality and specificity of the natural language prompts. The paper does not extensively discuss how sensitive the model is to variations in the prompts or provide strategies for handling ambiguous or incorrect prompts.

4. Although the authors conduct some ablation studies (e.g., the impact of hallucination quantification and text prompts), more detailed analyses could be beneficial. For instance, exploring how the number of experts affects performance (increasing the expert number for each modality), or how the model behaves when faced with unseen prompts or modalities, would provide deeper insights.

5. Some visualizations are not distinct in Figure 4, such as BUSI.

**Questions:**

Please see the weakness part 1,2,3,4,5.

---

### Official Review · Reviewer_E8bB · 2024-11-03

**Soundness:** 2
**Presentation:** 3
**Contribution:** 2
**Rating:** 5
**Confidence:** 5

**Summary:**

This paper presents a generalized anomaly detection approach for medical images. By leveraging visual-language models for text prompting, the proposed method extends reconstruction-based anomaly detection to multi-modality medical images. To address the issue of hallucinatory anomalies in normal images, a hallucination-aware loss function is introduced, which helps reduce false positives. The method’s effectiveness is demonstrated through experiments on five medical datasets, showing its superiority over prior approaches.

**Strengths:**

1. This paper tackles a crucial topic in medical image analysis. Anomaly detection plays a vital role in identifying rare diseases in medical imaging, and the generalized approach of the proposed method enhances its scalability.
2. The paper is well-written and easy to follow.
3. The hallucination-aware normalization sounds interesting and the corresponding ablation shows its effectiveness.
4. Extensive experiments are conducted, with ablation studies that effectively demonstrate the contributions of each component in the proposed method.

**Weaknesses:**

1. My main concern with this paper is the limited methodological innovation. The proposed approach uses a shared encoder with multiple decoders for anomaly detection, and the loss function optimizes the routing network to classify the domain of query images, selecting the appropriate decoder for reconstruction-based anomaly detection. This is also demonstrated in the ablation study in Fig. 6. From a methodological perspective, this setup is effectively equivalent to building several task-specific anomaly detection models that share a single encoder. Consequently, I find limited improvement in scalability, contrary to the claims made in the paper.
2. The comparative experiment in Table 1 is incomplete. According to [1], PatchCore[2] and reverse distillation[3] are two methods that generally perform well across multiple medical image anomaly detection tasks, yet they are missing from the single-task anomaly detection comparisons. More importantly, the solution in [4] represents the state-of-the-art for visual-language model-based medical image anomaly detection. Please include these SOTA methods in the experiments for a more comprehensive comparison.
3. For anomaly detection tasks, especially in medical imaging, accurately localizing anomaly regions is critical. Anomaly localization has become a standard measure to compare anomaly detection methods in prior studies, yet this evaluation is missing from the paper. What is the proposed method’s performance in terms of anomaly localization?

ref:
[1] Bao, Jinan, et al. "Bmad: Benchmarks for medical anomaly detection." Proceedings of the IEEE/CVF Conference on Computer Vision and Pattern Recognition (CVPR), 2024.
[2] Roth, Karsten, et al. "Towards total recall in industrial anomaly detection." Proceedings of the IEEE/CVF conference on computer vision and pattern recognition (CVPR), 2022.
[3] Deng, Hanqiu, and Xingyu Li. "Anomaly detection via reverse distillation from one-class embedding." Proceedings of the IEEE/CVF conference on computer vision and pattern recognition (CVPR), 2022.
[4] Huang, Chaoqin, et al. "Adapting visual-language models for generalizable anomaly detection in medical images." Proceedings of the IEEE/CVF Conference on Computer Vision and Pattern Recognition (CVPR), 2024.

**Questions:**

Please refer to the weakness section for my questions and comments.

---

### Note · Authors · 2024-11-15

I have read and agree with the venue's withdrawal policy on behalf of myself and my co-authors.